# Biocomposite Hydrogels for the Treatment of Bacterial Infections: Physicochemical Characterization and In Vitro Assessment

**DOI:** 10.3390/pharmaceutics13122079

**Published:** 2021-12-04

**Authors:** Delia Mihaela Rata, Anca Niculina Cadinoiu, Marcel Popa, Leonard Ionut Atanase, Oana Maria Daraba, Irina Popescu, Laura Ecaterina Romila, Daniela Luminita Ichim

**Affiliations:** 1Faculty of Medical Dentistry, Apollonia University of Iasi, 700511 Iasi, Romania; delia.rata@univapollonia.ro (D.M.R.); marpopa@tuiasi.ro (M.P.); leonard.atanase@univapollonia.ro (L.I.A.); oana.daraba@univapollonia.ro (O.M.D.); laura.romila@univapollonia.ro (L.E.R.); daniela.ichim@univapollonia.ro (D.L.I.); 2Petru Poni Institute of Macromolecular Chemistry, Aleea Grigore Ghica Voda 41A, 700487 Iasi, Romania; ipopescu@icmpp.ro

**Keywords:** chitosan, poly (vinyl alcohol), ZnO nanoparticles, hydrogels, antimicrobial activities

## Abstract

Hydrogels based on natural and synthetic polymers and inorganic nanoparticles proved to be a viable strategy in the fight against some Gram-positive and Gram-negative bacteria. Additionally, numerous studies have demonstrated the advantages of using ZnO nanoparticles in medicine due to their high antibacterial efficacy and relatively low cost. Consequently, the purpose of our study was to incorporate ZnO nanoparticles into chitosan/poly (vinyl alcohol)-based hydrogels in order to obtain a biocomposite with antimicrobial properties. These biocomposite hydrogels, prepared by a double crosslinking (covalent and ionic) were characterized from a structural, morphological, swelling degree, and mechanical point of view. FTIR spectroscopy demonstrated both the apparition of new imine and acetal bonds due to covalent crosslinking and the presence of the sulfate group following ionic crosslinking. The morphology, swelling degree, and mechanical properties of the obtained hydrogels were influenced by both the degree of covalent crosslinking and the amount of ZnO nanoparticles incorporated. In vitro cytotoxicity assessment showed that hydrogels without ZnONPs are non-cytotoxic while the biocomposite hydrogels are weak (with 3% ZnONPs) or moderately (with 4 and 5% ZnONPs) cytotoxic. Compared to nanoparticle-free hydrogels, the biocomposite hydrogels show significant antimicrobial activity against *S. aureus*, *E. coli*, and *K. pneumonia*.

## 1. Introduction

Bacterial infectious diseases represent a severe threat to human health and according to data from the World Health Organization (WHO), millions of patients around the world suffer from these bacterial infections each year [1]. The low effectiveness of antimicrobial agents on antibiotic-resistant bacterial infections can lead to increased mortality rates and prolonged hospitalization, but also to economic losses. Antibiotic abuse or incorrect use can lead to changes in the ability of microorganisms to resist antibacterial agents, leading to a decrease in their therapeutic efficacy [2,3]. Therefore, an increased number of researchers are involved in finding a viable strategy for the treatment of antibiotic-resistant bacterial infections by trying to explore new strategies without antibiotics or based on the administration of lower doses of drugs [1,4]. The development of new drug delivery systems that can extend the half-life, improve the bioavailability, optimize the pharmacokinetics, and decrease the dosing frequency of drugs may be a great solution to these issues [5]. Among various drug delivery systems, hydrogels can be a promising alternative due to their multiple advantages: nontoxicity, biocompatibility, biodegradability, mechanical properties, bioresorption ability, increased degree of flexibility, good permeability, easy loading, and controlled drug release but also due to their relatively low cost [6]. Hydrogels are three-dimensional hydrophilic polymer networks with a high capacity to absorb aqueous solutions or biological fluids, which have a degree of flexibility similar to natural tissue and that can generally be used for a wide range of bio-applications, such as drug delivery, artificial muscle manufacturing, enzyme and cell immobilization, contact lenses, biosensor membranes, artificial skin materials, and artificial heart liners [7,8]. Different types of hydrogels with potential antibacterial properties in the treatment of bacterial infectious (osteomyelitis [9], implant infections [10], skin wounds infectious [11], dental infectious, ocular infections [12]) were already reported in the literature.

Among all types of hydrogels, hydrogels from natural polymers (chitosan, dextran, gelatin, and hyaluronic acid) have great potential in medicine because of their biocompatibility, biodegradability, mechanical properties, bioresorption ability, and relatively low cost [13]. On the other hand, the synthetic polymers present a high chemical purity, reproducible synthesis process, controllable degradation rate, very good mechanical properties, and an important advantage to achieve a sustained release of the therapeutic agents over a longer period of time in comparison with the natural ones [14,15]. Hydrogels based on chitosan (CS) and poly (vinyl alcohol) (PVA) containing zinc oxide nanoparticles (ZnONPs) have already been studied in the literature by the freezing-thawing cycle [16,17] or solution casting method [18], but to the best of our knowledge, there are no studies on the obtaining of this type of biocomposite hydrogel using the double crosslinking method. In this context, the novelty of our study consists in the development of a new antibacterial biocomposite hydrogel based on CS and PVA containing ZnONPs using the double crosslinking method in order to reduce the quantity of the potential toxic covalent crosslinking agents.

Chitosan is a biocompatible, biodegradable, and non-toxic polysaccharide known to have antimicrobial properties and very commonly used to obtain various controlled release systems (hydrogels, nanoparticles, films, gels) with applications in medicine [19]. The main disadvantage of CS-based hydrogels is that they have low mechanical properties that restrict their practical application. However, in order to overcome this drawback, it is recommended to mix CS with synthetic polymers, such as PVA, which has good mechanical strength and is generally considered as a safe and biocompatible, non-toxic, and non-carcinogenic material that can be crosslinked by various physical and chemical methods to obtain hydrogels [20,21]. In our study, we used two crosslinkers for the preparation of hydrogels, the covalent one, glutaraldehyde (GA), and the ionic one, magnesium sulfate (MgSO_4_). In our previous studies, it was demonstrated that this double crosslinking is a viable strategy for the preparation of hydrogels in the form of films or micro/nanoparticles [22,23,24,25,26]. An important advantage of this strategy is that the amount of covalent crosslinker, which can be potentially toxic for biomedical applications, is reduced but it still allows hydrogels with good mechanical properties to be obtained. Moreover, it is well known that the incorporation of inorganic antibacterial materials into hydrogels leads to the development of complex systems with superior mechanical strength, increases adhesion to surfaces (skin, soft tissues), and enhances antibacterial properties [27,28]. ZnONPs have been shown to possess great antibacterial activities on broad-spectrum pathogenic bacteria (*Staphylococcus aureus*, *Bacillus subtilis*, *Escherichia coli*, *Salmonella enteritidis*, *Salmonella typhimurium*, *Pseudomonas fluorescens, Pseudomonas aeruginosa*) [29,30,31]. The production of reactive oxygen species, loss of cellular integrity as a result of contact between ZnONPs and the cell wall, internalization of ZnONPs, release of Zn^2+^ ions, and photoconductivity are the most important antimicrobial mechanisms responsible for the destruction of microorganisms [32].

The urgent need to combat bacterial resistance to antibiotics has led to an increase in the research of alternative ways to control pathogenic attacks. Therefore, biocomposite hydrogels were obtained in this study and their physicochemical, morphological, and mechanical characterization and the in vitro cytotoxicity and antimicrobial activity against a variety of microorganisms that included Gram-positive bacteria (*Staphylococcus aureus*) and Gram-negative bacteria (*Escherichia coli*, *Klebsiella pneumoniae*, and *Pseudomonas aeruginosa*) were evaluated.

## 2. Materials and Methods

### 2.1. Materials

Medium molecular weight CS, degree of deacetylation~75% (molecular weight: 100,000–300,000 g/mol), and PVA, 13,000–23,000 g/mol, 87–89% degree of hydrolysis were purchased from Acros Organics BVBA (Geel, Belgium). Glutaraldehyde (GA) 50% in aqueous solution, anhydrous ≥98.0% magnesium sulfate (MgSO_4_), and glycerine ≥99.7% were purchased from VWR International. Zinc oxide nanopowder (ZnO, 99+%, 10–30 nm) was obtained from US Research Nanomaterials, Inc. (Houston, TX, USA). Human dermal fibroblasts cell line (HDFa), Dulbeco’s modified Eagle’s Medium (DMEM), 10% fetal bovine serum (FBS), antibiotics (streptomycin/penicillin), non-essential amino acids, phosphate-buffered saline (PBS), and trypsin-EDTA, necessary for in vitro cytotoxicity assay, were purchased from Thermo Fisher Scientific (Waltham, MA, USA). 3-(4,5-Dimethyl-2-thia zolyl)-2,5-diphenyl-2H-tetrazolium bromide was obtained from Merck Millipore (Darmstadt, Germany). Freeze-dried stains (*Staphylococcus aureus*-ATCC 25923, *Escherichia coli*-ATCC 11775, *Klebsiella pneumonia*-ATCC BAA–1705, and *Pseudomonas aeruginosa*-ATCC 10145) were purchased from ATCC (Manassas, VA, USA). Chapman agar (mannitol salt agar) was purchased from Oxoid (Hampshire, United Kingdom) and MacConkey agar from G & M Procter Ltd. (Perth, UK). All other chemicals used were of analytical grade.

### 2.2. The Preparation of Biocomposite Hydrogels

Double crosslinked hydrogels were obtained according to the experimental program provided in Table 1.

Appropriate amounts of CS and PVA were dissolved separately, CS in 25 mL of lactic acid solution 1% (*w*/*v*) and PVA in 25 mL of ultrapure water. After their complete dissolution, the two solutions were mixed. The specific amount of GA required to crosslink the free amine of CS and hydroxyl groups of PVA was added dropwise under vigorous stirring over the mixed polymer solution. After 30 min, the MgSO_4_ solution was added, stirring for a further 15 min. In order to increase the elasticity of hydrogels, 500 mg of glycerine were added to the polymer solution and the mixture was to stirred for another 15 min. After that, the obtained mixture was carefully transferred into a square silicone mold with a side of 6.5 cm. The samples were dried in the oven at 37 °C. After purification (washing with ultrapure water), the hydrogels were stored in the desiccator, at a constant humidity of 65%.

To impart antimicrobial properties to hydrogels, ZnO nanoparticles were added to their composition. Specific amounts of inorganic nanoparticles (Table 1) were first added to the PVA solution and allowed to stir overnight, at room temperature and in the dark. The next day, CS was added to this mixture and the same steps, as described above, were followed in order to prepare biocomposite hydrogels.

### 2.3. Characterization of Biocomposite Hydrogels

#### 2.3.1. The Structural Characteristics

The structural characteristics of the obtained hydrogels, the ionic and covalent new bonds, and also the presence of ZnONPs were analyzed using a Bruker FT-IR Spectrometer (VERTEX 70) equipped with a DLaTGS detector in the ATR operating mode (ZnSe crystal: 600–4000 cm^−1^). 

#### 2.3.2. The Morphological Characteristics

The morphological characteristics of the hydrogels were determined by scanning electron microscopy (SEM) using a Quanta 200 Scanning Electron Microscope (FEI Company, Bruno, Czech Republic). The hydrogels were analyzed in the dry state. This analysis gave us information on the porosity and homogeneity of hydrogels. 

The shape and size of ZnONPs were assessed using a HITACHI-HT7700 Transmission Electron Microscope (Hitachi High-Technologies Corporation, Tokyo, Japan). 

The elemental composition of the hydrogels and the presence of ZnONPs were demonstrated using a Verios G4 UC Scanning Electron Microscope from Thermo Scientific (Bruno, Czech Republic) equipped with an Octane Elect Super SDD detector (an energy-dispersive X-ray spectrometer from EDAX (Ametek, Mahwah, NJ, USA).

To provide electrical conductivity and to prevent charge buildup during exposure to the electron beam, the samples were coated with 6 nm platinum using a Leica EM ACE200 Sputter coater.

#### 2.3.3. The Swelling Behavior

The swelling behavior of the obtained hydrogels was evaluated, using the gravimetric method, at 37 °C, in phosphate buffer solution (PBS) at pH 7.4, which mimics biological fluids [22]. The dried square-shaped hydrogel samples (1 cm side) were first weighed (W_d_, mg) and then immersed in 20 mL of PBS. At predetermined times, the samples were easily removed from PBS and placed on a filter paper. The excess PBS on the surface of the hydrogels was easily absorbed with another filter paper and the swollen samples were weighed (W_st_). This procedure was repeated until equilibrium was reached. The swelling degree at time t (*Q_t_*(%)) was calculated using the following equation:(1)$$Q_{t}\left(\%\right)=\frac{W_{\mathrm{st}}-W_{d}}{W_{d}}\times 100\;$$

#### 2.3.4. Tensile Mechanical Measurements

Tensile mechanical measurements were performed using a Brookfield CT3 Texture Analyzer (Texture Brookfield Engineering Laboratories Inc., Middleboro, MA, USA) with a 50N load cell, at room temperature. The Roller Cam Accessory (TA-RCA) was used for this test. The films (dry hydrogels kept in an atmosphere with 65% humidity) were cut into rectangles (length 55 mm, width 10 mm) and were caught between two clamps positioned at a distance of 30 mm. The thickness of the films was determined by taking the average of the thickness, measured at five different places with a digital Caliper and found to be around 0.5 mm in all the samples. The speed of the tensile measurements was fixed at 0.5 mm/s. The stress (σ, N/m^2^) was calculated from the load divided by the cross-sectional area of the undeformed sample, and the strain (ε) was determined as the clamp displacement divided by the initial distance between the two clamps. The ultimate strain and stress (from the point of failure) were determined, and the elastic modulus values were calculated from the slope of the linear climbing tract of the stress–strain plot within the fixed strain region 0.1–2%.

#### 2.3.5. The In Vitro Cytotoxic Effects

The in vitro cytotoxic effects of the hydrogels were assessed using human dermal fibroblasts, adult, and cell line (HDFa). HDFa cells were cultured and prepared for testing as described in our previous paper [33]. The sterilized hydrogel samples were cut with a biopsy punch to obtain discs 4 mm in diameter and incubated with the HDFa cells. After the incubation time (24 and 48 h), the cell viability was determined using the MTT assay. All procedures were performed in a laminar flow hood (Lamil Plus 13, Kartusalan Metally Oy). Each hydrogel sample was tested in triplicate in order to get the average percentage and the standard error. 

#### 2.3.6. The Antimicrobial Activities

The antimicrobial activities of the obtained hydrogels without and with ZnONPs were evaluated using the disk diffusion assay against 4 reference strains: *S. aureus* (Gram-positive bacteria), *E. coli*, *K. pneumoniae*, and *P. aeruginosa* (Gram-negative bacteria). Commercially available antimicrobial disks with Chapman agar for Gram-positive bacteria, and MacConkey agar for Gram-negative bacteria were used for these tests. A suspension of microorganisms (0.5 McFarland density) was inoculated on a Petri dish with the substrate. The sterilized hydrogel discs (4 mm diameter) were firstly hydrated in sterile 0.9% saline solution and then placed on the agar plate and incubated for 24 h at 37 °C. The antimicrobial activities were evaluated by measuring the diameters of the inhibition zones (din, mm) according to the Kirby–Bauer method [34]. The experiments were repeated three times. In order to evaluate the significance of their antibacterial activity against *S. aureus*, *E. coli*, and *K. pneumonia* strains, as compared to samples without ZnONPs, the one-way ANOVA statistical test was processed.

## 3. Results and Discussions

### 3.1. Preparation of Biocomposite Hydrogels

Double crosslinked biocomposite hydrogels, based on CS and PVA, with antimicrobial properties were obtained by incorporating of ZnONPs into the polymer solution before crosslinking. The preparation method and the structure of the obtained biocomposite hydrogels is represented schematically in Figure 1. The reaction with GA takes place both at the amino groups from the CS chain, with the formation of imine bonds, and at the hydroxyl groups from PVA with the formation of new acetal-type bonds. Ionic crosslinking was performed using MgSO_4_. Amino-type functional groups in the form of the ammonium ion (belonging to CS) with the sulfate groups of the ionic crosslinking agent participate in this reaction.

The optimization of the obtaining conditions and the addition of the plasticizer (glycerine) led to the obtaining of hydrogels with improved mechanical properties. Additionally, the addition of ZnO nanoparticles to the polymer mixture yielded new biocomposite hydrogels with antimicrobial properties. 

### 3.2. Characterization of Biocomposite Hydrogels

#### 3.2.1. The Structural Characteristics

The structural characteristics of the obtained hydrogels were determined using FT-IR spectroscopy. The FT-IR spectra of hydrogel samples with different degrees of covalent crosslinking in the absence and in the presence of ZnONPs are shown in Figure 2 and Figure 3, respectively.

The obtained results confirmed, on the one hand, the reaction between the amino functional groups from CS and the hydroxyl functional groups of PVA with the covalent crosslinker (GA), and on the other hand, the reaction between the amino functional groups from CS with the SO4^2−^ groups of the ionic crosslinker (MgSO_4_). The characteristic peaks located around 3314, 3310, 3312, and 3303 cm^−^^1^, respectively, that were found in the spectrum from Figure 2, and peaks located between 3305 and 3323 cm^−^^1^ in the spectrum from Figure 3 can be attributed to vibration stretching of secondary amine groups (NH) from CS or hydroxyl groups (OH) from PVA that are involved in the intra- and intermolecular bonds of hydrogen [35,36]. The absorption bands at about 2849 cm^−1^ and those between 2920 and 2940 cm^−1^ (Figure 2 and Figure 3) correspond to the C-H bonds in the alkyl groups. In the case of samples with a higher amount of GA (C2P2.15, C2P4.15, C2P4.15.Z3, and C2P4.15.Z5), there is an intensification of the peak observed at 2849 cm^−1^. Absorption bands around 1732 cm^−1^, which are found in the spectra of all analyzed samples, can be associated with the stretching vibration of the —C=O group in PVA [23]. The peaks noticed for all the analyzed samples at 1629–1646 cm^−1^ can be attributed to the imine group (C=N), which is proof of the covalent crosslinking between GA and CS. The intense peak located at approximately 1590 cm^−^^1^ indicates the presence of amine groups (NH_2_) specific for a polysaccharide structure [24]. The absorption bands at 1035–1040 and 1082–1088 cm^−1^ can be attributed to the acetal groups (C-O-C) formed by crosslinking of PVA with GA. The presence of ZnONPs into hydrogels was confirmed by the appearance of the peak at approximately 774 cm^−1^, which corresponds to the ZnO stretching vibration [37]. The absorption peaks ranging from approximately 654 to 667 cm^−1^ that are found in the spectra of all analyzed samples can be attributed to the SO4^2−^ group, which evidenced the ionic crosslinking between the functional groups of CS with MgSO_4_ [25].

#### 3.2.2. Morphological Characteristics

##### ZnO Nanoparticles

Figure 4A shows the transmission electron microscopy (TEM) image of the ZnONPs. The TEM micrographs revealed that these nanoparticles, with a hexagonal shape, have dimensions of approximately 30 nm in a dry state.

EDAX elemental analysis of ZnONPs (Figure 4B) showed a large amount of Zn of approximately 73%. The amount of O present in the EDAX spectrum was approximately of 18%.

##### Biocomposite Hydrogels

The cross-section morphology of the hydrogel samples, with and without ZnONPs, was studied by scanning electron microscopy (SEM). The micrographs presented in Figure 5 reveal a fibrillar structure probably due to the presence of polysaccharide (CS) in the sample composition. It was also observed that the hydrogels become smoother when the amount of covalent crosslinker increases. Increasing the amount of covalent crosslinker increases the crosslinking density because an increased number of imine and acetal groups are formed, and this makes the structure of hydrogels smoother. This is in accordance with the swelling degree results, which reveal that as the amount of covalent crosslinker increases, the degree of swelling decreases. The hydrogels have a homogeneous structure, due to the strong covalent and ionic crosslinking, and the samples without ZnONPs did not show porosity. On the contrary, it has been observed that the hydrogels obtained in the presence of ZnONPs shows porosity, with the pores being small and irregular in size. At this point, it is very important to mention that with the increase of the amount of ZnONPs, a higher porosity was observed. Following the purifications, a part of the ZnONPs was removed and the place occupied by these nanoparticles inside the hydrogel matrix led to the appearance of pores and implicitly to the increase of the porosity. This effect is also illustrated by the results of the EDAX tests.

##### EDAX Analysis for Biocomposite Hydrogels

EDAX elemental analysis revealed the presence of ZnONPs in the analyzed biocomposite hydrogels (Figure 6). 

The EDAX images also illustrated the presence of large amounts of C and O. The final content of ZnONPs in the tested hydrogels is presented in Table 2.

Following the calculations performed based on the EDAX analysis (Figure 4B, Table 2 and Appendix A), it was found that the remaining percentages of ZnNPs in biocomposite hydrogels were approximately 70% in C2P4.10.Z3 hydrogel, approximately 93% in C2P4.10.Z5 hydrogel, approximately 71% in C2P4.15.Z3 hydrogel, and approximately 95% in C2P4.15.Z5 hydrogel. The obtained results demonstrate that the percentage of ZnONPs in the system is higher when the covalent crosslinking density increases (the amount of GA increases). After the purification step, a small amount of ZnONPs was removed from the biocomposite hydrogel network, thus explaining the decrease in the percentage of ZnONPs in the hydrogel compared to the initial one. The elemental composition of all tested samples is given in the Appendix A.

#### 3.2.3. Swelling Behavior

The swelling behavior of the obtained hydrogels was investigated in slightly alkaline medium (PBS at pH 7.4) in order to evaluate their potential to be used locally in biomedical applications for the treatment of bacterial infections. The swelling of the hydrogels was caused by the penetration of the aqueous solution, by diffusion, into the meshes of the polymer network. The amount of aqueous solution penetrating inside the hydrogel depends on the membrane elasticity, which can influence the diffusion rate. The swelling behavior of hydrogels was influenced by both the amount of covalent crosslinking density (Figure 7A) and the amount of ZnONPs (Figure 7B). As expected, it was found that the increase of the amount of covalent crosslinker (increase of the ratios between GA and the functional groups of the two polymers) led to a decrease of the swelling degree (Figure 7A). This behavior can be attributed to the increase of the crosslinking density. Additionally, increasing the amount of ZnONPs in hydrogels also leads to a decrease of the swelling degree (Figure 7B). This behavior can be attributed to the decrease of the space inside the hydrogel network due to the increase of the amount of ZnONPs present in the system. Therefore, the amount of water that penetrated the hydrogel network was reduced.

It should also be mentioned that after 6 h, all the samples reached the maximum swelling degree. In fact, between 6 and 24 h, there were no changes in the samples’ swelling degree as can be seen in Figure 7.

#### 3.2.4. Tensile Mechanical Measurements

The mechanical properties of hydrogels are one of most important factors for establishing their future biomedical applications. Tensile properties of the dry hydrogels/films were investigated, and the obtained stress–strain curves are presented in Figure 8A,B. The tensile strength and modulus of elasticity are plotted in Figure 8C,D. Glycerine, a typical plasticizer for hydrophilic films, was used to obtain films that are not brittle, with an elongation at break between 20 and 50%. The tensile strength ranged from 2.3 to 5.6 MPa and the elastic modulus from 6 to 30 MPa, but the composition and the crosslinking density influenced the mechanical properties of the hydrogel network.

The increase of the crosslinking leads to a decrease in the free volume between the chains, with a consequent stiffening of the network. Therefore, the crosslinking determines an enhancement in the tensile strength and Young’s modulus and a decrease of the elongation at break [38,39,40,41]. This behavior was also observed for the double crosslinked PVA/CS hydrogels: the modulus of elasticity and tensile strength increased in the C2P2 series (Figure 8C) and C2P4 series (Figure 8D) with the increase of the GA content. The increase of the PVA content in the C2P2 to C2P4 samples determined an increase of the tensile strength and Young’s modulus, which is due to the intermolecular interaction between PVA and CS chains through hydrogen bonds [42,43] but can also be explained by the higher chemical crosslinking of the C2P4 networks compared to C2P2 networks. 

When ZnONPs were incorporated into the hydrogels, the tensile strength and elongation at break decreased compared with the sample without metal oxide NPs (Figure 8B,D). A higher effect was observed for the C2P4.15.Z3 sample, but with a further increase of the ZnONPs concentration, the mechanical properties were improved. It is known that the addition of metallic oxide nanofillers in CS films generally improved their mechanical properties [44,45], but in this case, ZnONPs probably interfere with the ionic crosslinking. However, the hydrogels with ZnONPs had a lower modulus of elasticity, compared with C2P4.15. The higher elasticity of the network is attributed to the fact that ZnONPs weakened the intermolecular hydrogen bonds between CS chains or between CS and PVA [45,46].

After analyzing the obtained results, two of the samples with the good mechanical properties, C2P4.10 and C2P4.15, were selected for subsequent cytotoxicity and antimicrobial testing. 

#### 3.2.5. The In Vitro Cytotoxic Effects

Cytotoxicity tests provide necessary information in order to evaluate the biocompatibility of materials that will be used in biomedical applications. For in vitro cytotoxicity assessment of hydrogels (in the absence and in the presence of ZnONPs), human dermal fibroblast cells (HDFa) were used as model cells. The effect of hydrogels on the viability of fibroblasts after incubation periods of 24 and 48 h was evaluated using the MTT colorimetric assay and the obtained results are shown in Figure 9. The MTT assay was adapted according to the protocol described by Mosmann [47] and calculates cell viability based on mitochondrial function by reducing 3-(4,5-dimethylthiazol-2-yl)-2,5-diphenyltetrazolium bromide (MTT) to a colored insoluble formazane salt.

As can be seen from Figure 9, the viability of cells in contact with hydrogels without ZnONPs was around 89% after 24 h and around 84% after 48 h, which proves that they are non-cytotoxic [48,49]. The addition of an increasing amount of ZnONPs in hydrogels has led to a decrease in cell viability. In the case of hydrogels with ZnONPs, cell viability after 24 h of incubation was between 62% and 78%, demonstrating that these samples show a weak cytotoxicity. After 48 h of incubation, there was a decrease in cell viability, with only the samples with 3% ZnONPs remaining in the weak cytotoxic range, while the others decreased in the moderate cytotoxic range. A small decrease in viability was observed when the amount of covalent crosslinker (GA), used to obtain hydrogels, was increased. 

There are studies in the literature that have evaluated the cytotoxic effect of ZnONPs and found that they are toxic to several cell types [50,51]. Therefore, their incorporation into a hydrogel matrix is a solution to reducing their cytotoxic effect, as it was concluded by different studies where hydrogels with ZnONPs based on bacterial cellulose (CS) [52] or sodium alginate [53] were tested.

#### 3.2.6. The Antimicrobial Activities

Agar disk-diffusion testing is the method used in some clinical microbiology laboratories for routine antimicrobial susceptibility testing [54]. This method was also used in this study in order to evaluate the antimicrobial activity for the hydrogels based on CS and PVA, obtained in the absence and in the presence of ZnONPs. Typical reference strains of *S. aureus* (Gram-positive bacteria) and *E. coli*, *K. pneumoniae*, and *P. aeruginosa* (Gram-negative bacteria) were used for these tests (Figure 10). Four samples were placed on each agar plate: the hydrogels without ZnONPs (A-C2P4.10 or B-C2P4.15) and three hydrogel samples with different concentrations of ZnONPs (3% ZnONPs for E-C2P4.10.Z3 or G-C2P4.15.Z3; 4% ZnONPs for F-C2P4.10.Z4 or H-C2P4.15.Z4; 5% ZnONPs for C-C2P4.10.Z5 or D-C2P4.15.Z5).

The diameter of the inhibition zone (d_iz_), obtained after 24 h of incubation at 37 °C, was transcribed as “−” when no antimicrobial effect was noticed, “+” when the d_iz_ is <15 mm, and “++” when the d_iz_ is between 15 and 25 mm [55]. These results are presented in Figure 10.

As CS and ZnONPs have both antibacterial properties, a synergistic effect might be expected. Antibacterial mechanisms for both materials have been previously reported in the literature [56,57,58,59]. Consequently, hydrogel samples containing ZnONPs show significant antibacterial properties, while samples without ZnONPs have no detectable antimicrobial activity (Figure 11). As the concentration of ZnONPs increases, the zone of inhibition also increases. Therefore, the highest antimicrobial activity was observed for samples C2P4.10.Z5 (C) and C2P4.15.Z5 (D) (hydrogels with 5% ZnONPs) against *S. aureus*. These hydrogel samples also showed an antimicrobial effect against *E. coli* and *K. pneumonia.* The diameter of the inhibition zone was smaller in the case of samples C2P4.10.Z4 (F) and C2P4.15.Z4 (H) (hydrogels with 4% ZnONPs), with these having an antimicrobial effect against *E. coli.* Sample H, which was obtained with a higher amount of covalent crosslinker, also has antimicrobial activity against *S. aureus*. However, it appeared that no hydrogel sample showed antimicrobial activity on the *P. aeruginosa* strain. Similar results were observed in studies conducted by other researchers [29].

The inhibition zone diameter data from the antibacterial tests for the samples containing 4 and 5% ZnONPs were processed through the one-way ANOVA statistical test in order to evaluate the significance of their antibacterial activity against *S. aureus* (Gram-positive bacteria), *E. coli*, and *K. pneumonia* (Gram-negative bacteria) strains, as compared to the samples without ZnONPs (Appendix A).

Since, for all bacteria strains, the significance level is lower than 0.05, it can be stated that the samples containing 4 and 5% ZnONPs have a significant influence on the inhibition of the growth of tested bacteria.

## 4. Conclusions

Within this study, a new attempt was made in order to obtain double crosslinked hydrogels, based on CS and PVA with embedded ZnONPs, for the potential treatment of bacterial infections. Both the successful crosslinking of the two polymers and the presence of ZnONPs in the biocomposite hydrogel structure were demonstrated by FT-IR spectroscopy. The morphology, swelling, mechanical properties, cytotoxicity, and antimicrobial effect were influenced by the covalent crosslinking degree and the amount of ZnONPs incorporated into hydrogels. The porosity of the hydrogels increased with an increase of the concentration of ZnONPs. On the contrary, the swelling degree decreased when the concentration of the ZnONPs increased. A similar trend was noticed for the cytotoxicity tests after 24 and 48 h. In vitro cytotoxicity tests have shown that only hydrogels without ZnONPs are non-toxic to fibroblast cells. The biocomposite hydrogels proved to be weak (with 3% ZnONPs) and moderately (with 4 and 5% ZnONPs) cytotoxic. 

The increased concentration of GA led to increased mechanical properties, but when ZnONPs were incorporated into the hydrogels, the tensile strength and elongation at break decreased compared with the sample without metal oxide NPs. The higher effect was observed for the C2P4.15.Z3 sample, but with a further increase of the ZnONPs concentration, the mechanical properties were improved. However, the hydrogels with ZnONPs had a lower modulus of elasticity, compared with the sample without metal oxide NPs.

In contrast to samples without ZnONPs, hydrogel samples containing 4 and 5% ZnONPs showed significant antibacterial activity against *S. aureus* (Gram-positive bacteria), *E. coli*, and *K. pneumonia* (Gram-negative bacteria) strains with an inhibition zone between 15 and 20 mm. The ANOVA test showed that these results are significant (significant level < 0.05). 

The obtained results are promising and therefore the obtained biocomposite hydrogels will be further investigated by in vivo analyses. 

## Figures and Tables

**Figure 1 pharmaceutics-13-02079-f001:**
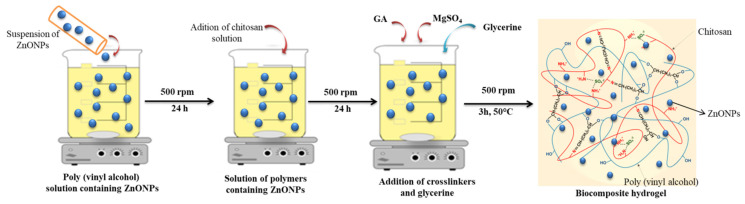
The schematic representation of the preparation method and the structure of the obtained biocomposite hydrogel.

**Figure 2 pharmaceutics-13-02079-f002:**
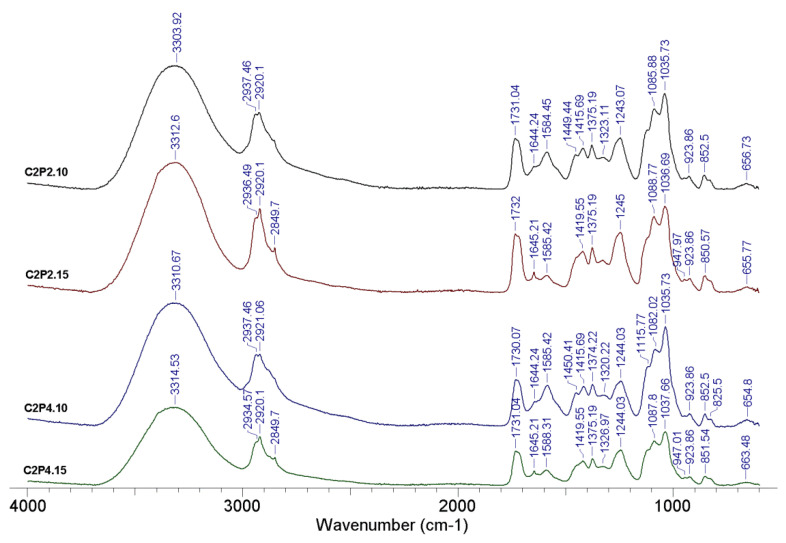
FT-IR spectra for hydrogel samples with different amounts of GA.

**Figure 3 pharmaceutics-13-02079-f003:**
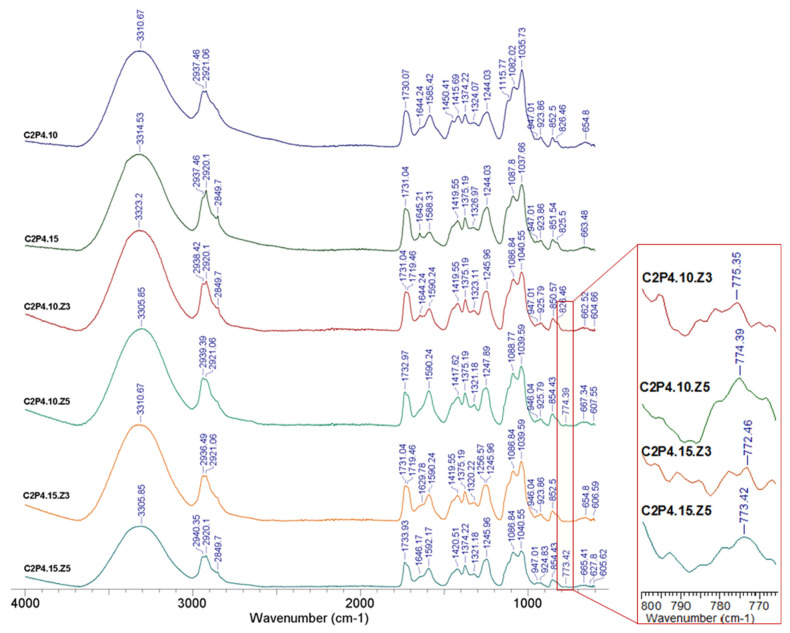
FT-IR spectra for hydrogel samples in the absence and in the presence of ZnONPs and different degrees of crosslinking.

**Figure 4 pharmaceutics-13-02079-f004:**
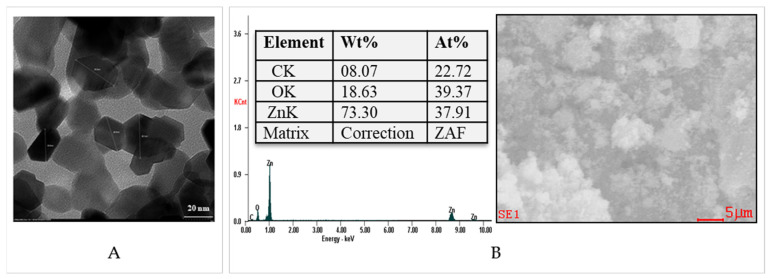
Transmission electron microscopy (TEM) (**A**) and EDAX spectrum (**B**) for ZnONPs.

**Figure 5 pharmaceutics-13-02079-f005:**
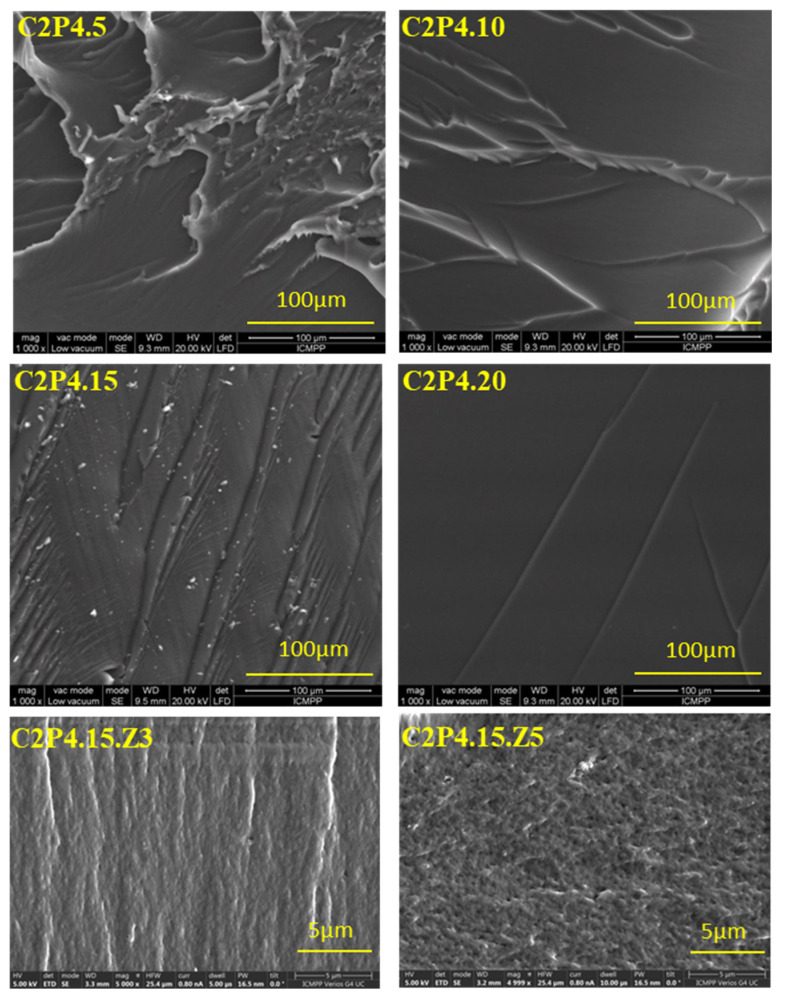
SEM micrographs for hydrogel samples in the cross-section with and without ZnONPs.

**Figure 6 pharmaceutics-13-02079-f006:**
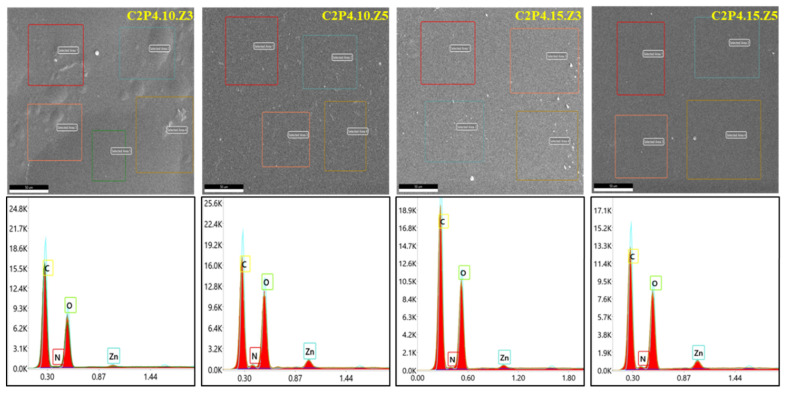
EDAX spectrum for samples C2P4.10.Z3, C2P4.10.Z5, C2P4.15.Z3, and C2P4.15.Z5.

**Figure 7 pharmaceutics-13-02079-f007:**
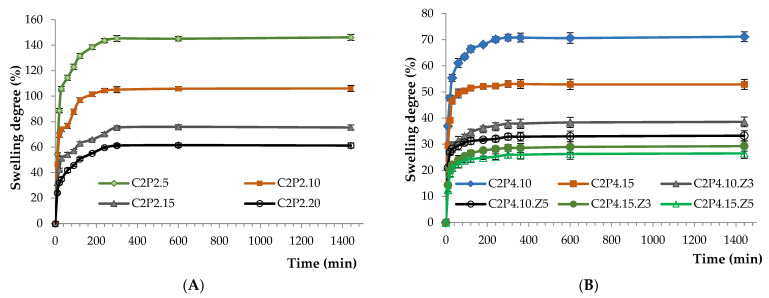
Swelling degree kinetics for 24 h in PBS (pH = 7.4) of (**A**) C2P2 hydrogels with different ratios between GA and the functional groups of the two polymers; (**B**) C2P4 hydrogels obtained in the absence and in the presence of ZnONPs.

**Figure 8 pharmaceutics-13-02079-f008:**
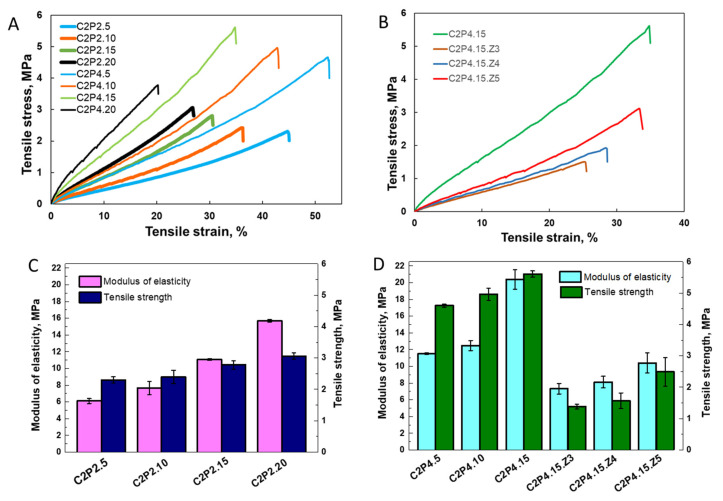
Tensile stress–strain curves for C2P2 and C2P4 series (**A**) and for C2P4.15 sample with and without ZnONPs (**B**). Variation of the modulus of elasticity and tensile strength for the C2P2 series (**C**) and C2P4 series (**D**).

**Figure 9 pharmaceutics-13-02079-f009:**
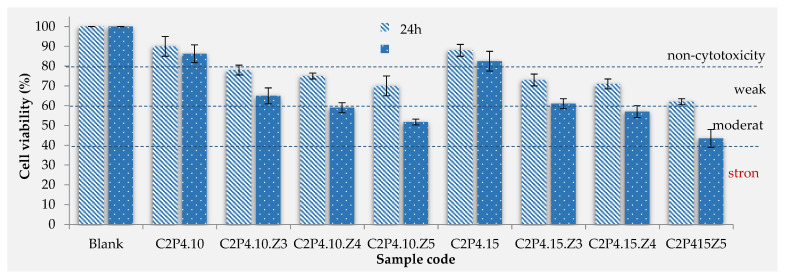
In vitro cell viability of C2P4.10 and C2P4.15 hydrogels obtained in the absence and in the presence of ZnONPs at 24 and 48 h after incubation.

**Figure 10 pharmaceutics-13-02079-f010:**
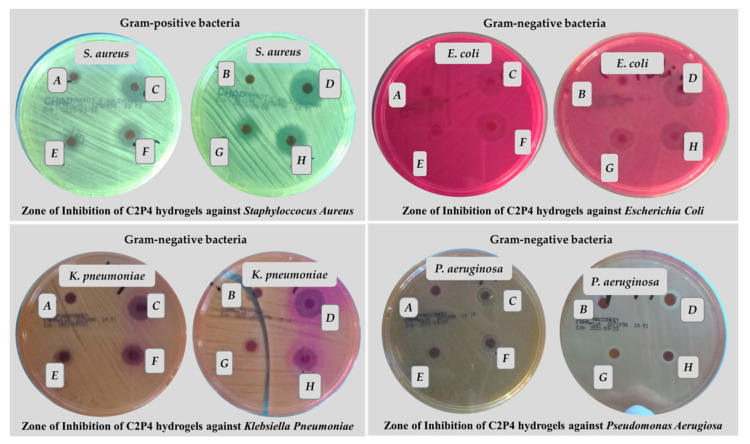
Antibacterial activity of C2P4.10 (A), C2P4.15 (B), C2P4.10.Z5 (C), C2P4.15.Z5 (D), C2P4.10.Z3 (E), C2P4.10.Z4 (F), C2P4.15.Z3 (G), and C2P4.15.Z4 (H) hydrogels against Gram-positive (*S. aureus*) and Gram-negative bacteria (*E. coli, K. pneumonia*, and *P. aeruginosa*).

**Figure 11 pharmaceutics-13-02079-f011:**
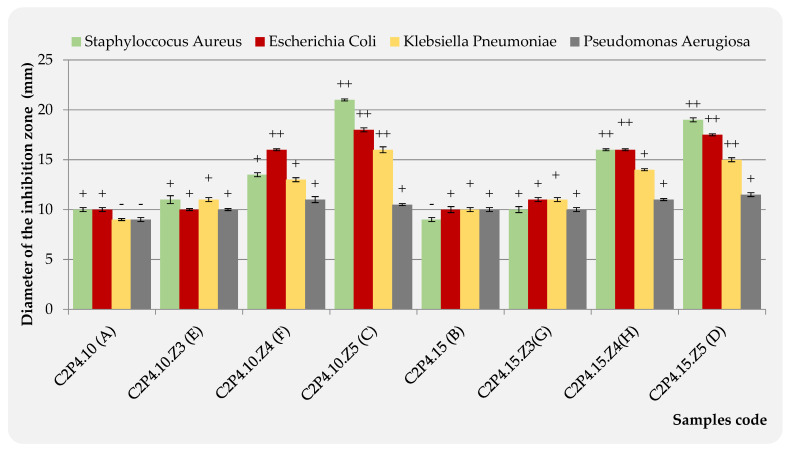
Antimicrobial effect of C2P4-10 (A), C2P4-15 (B), C2P4-10.Z5 (C), C2P4-15.Z5 (D), C2P4-10.Z3 (E), C2P4-10.Z4 (F), C2P4-15.Z3 (G), and C2P4-15.Z4 (H) hydrogels against *S. aureus*, *E. coli, P. aeruginosa*, and *K. pneumoniae*.

**Table 1 pharmaceutics-13-02079-t001:** The experimental design for double crosslinked hydrogels preparation.

Sample	CS/PVA Ratio (mg/mg)	GA/Free NH_2_ and OH Ratio (mol/mol)	MgSO_4_ / Free NH_2_ (mol/mol)	ZnO Nanoparticles in Relation to the Total Amount of Polymers (%)
C2P2.5	50/50	1/20	1/20	0
C2P2.10		1/10		
C2P2.15		3/20		
C2P2.20		1/5		
C2P4.5	50/100	1/20		
C2P4.10		1/10		0
C2P4.10.Z3				3
C2P4.10.Z4				4
C2P4.10.Z5				5
C2P4.15		3/20		0
C2P4.15.Z3				3
C2P4.15.Z4				4
C2P4.15.Z5				5
C2P4.20		1/5		0

**Table 2 pharmaceutics-13-02079-t002:** Content of ZnONPs in tested hydrogels.

Sample	C2P4.10	C2P4.10.Z3	C2P4.10.Z5	C2P4.15	C2P4.15.Z3	C2P4.15.Z5
Initial content of ZnONPs (*w*/*w* %)	0	3	5	0	3	5
Final content of ZnONPs (*w*/*w* %)	0	2.09	4.63	0	2.14	4.74

## Data Availability

The data presented in this study are available on request from the corresponding author.

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
