# Peer review of "Biocomposite Hydrogels for the Treatment of Bacterial Infections: Physicochemical Characterization and In Vitro Assessment"

_pharmaceutics, 2021, doi:10.3390/pharmaceutics13122079_

Round 1
Reviewer 1 Report
Biocompozite hydrogels for the treatment of bacterial infections: Physicochemical characterization and in vitro assessment is an interesting one and could be published after minor revision.
The new hydrogel proposed by the authors was prepared by double crosslinking and was characterized from structural, morphological, mechanical and antimicrobial point of view.
The style is clear and the experiments are well conducted. The conclusions are sustained by the experimental facts.
The obtained hydrogels seem to be a promising one for wound healing.
I suggest to use only one name for plasticizer glycerin or glycerol.
Author Response
Response to Reviewer 1 Comments
Point 1: Biocompozite hydrogels for the treatment of bacterial infections: Physicochemical characterization and in vitro assessment is an interesting one and could be published after minor revision.
The new hydrogel proposed by the authors was prepared by double crosslinking and was characterized from structural, morphological, mechanical and antimicrobial point of view.
The style is clear and the experiments are well conducted. The conclusions are sustained by the experimental facts.
The obtained hydrogels seem to be a promising one for wound healing.
Response 1: Thanks for your appreciation and suggestions.
Point 2: I suggest to use only one name for plasticizer glycerin or glycerol.
Response 2: We replaced glycerin and glycerol, both in the text and in Figure 1, with glycerine, such as the name of the plasticizer at the supplier.
Reviewer 2 Report
The manuscript authored by Delia Mihaela Rata et al. reports the synthesis of quitosan/poly(vinyl alcohol) hydrogels double crosslinked by covalent and ionic interactions, and containing Zn based nanoparticles for treatment of bacterial infections. The biological characterization is not very complete and there are several points that need to be addressed. I consider the manuscript does not meet the quality required for publication in Pharmaceutics, and a journal focused on Materials Science is recommended.
1) English should be strongly revised.
2) The authors should double checked and correct the typos in the title. It is important to replace “biocompozite” by “biocomposite”.
3) The authors should double checked and revise the abbreviations. Does GA mean glutaric aldehyde or glutaraldehyde? It is not mentioned in the same way along the manuscript.
4) In Materials section, reagents for biological tests, cytotoxicity and antimicrobial properties, need to be incorporated.
5) How the authors determine the specific amount of GA required to crosslink free amine of CS and PVA? (lines 119-120)
6) The Table 1 could be modified to in order to show a homogenous format. In the fourth column cells are combined when they show the same value. I suggest doing the same for the second and third columns.
7) Regarding morphological characterization of hydrogels by SEM, how samples were dried and prepared for SEM observation? Cryo-SEM measurements need to be incorporated in the work, because they are necessary to show the hydrogel structure avoiding the drying effect and collapsing of the hydrogel network.
8) Mechanical properties of hydrogels in wet state need to be determined. Rheological properties could be a very good option for that.
9) In vitro cytotoxicity at longer times is necessary to prove the non-cytotoxic effect along the time.
10) Figure 1 is confusing and it is not concordant with section 2.2 for hydrogels preparation. In the text, it is mentioned that CS is mixed with ZnONPs previously to mix with PVA, and in the scheme of Figure 1,it looks like that CS is mixed with PVA before ZnONPs addition.
11) The letter size of axes in Figures 2, 3, and 9 is not easily readable, and it should be increased.
12) The appearance of a peak around 774 cm-1 is not clear in Figure 3.
13) Figures 4 and 5 could be grouped together.
14) At what pH takes place the hydrogel formation? That is very important in the case of electrostatic interactions and for the swelling test in PBS at pH 7.4.
15) Why hydrogels are smoother when the covalent crosslinker concentration is increased?
16) Caption of Figure 7 should be double checked to show information of all samples.
17) If glycerin is incorporated in the tensile mechanical measurements, it should be incorporated in the hydrogels formulation to be characterized by the rest of morphological, swelling, biological, etc., techniques employed in this work.
18) Authors claim in the section 3.2.5 (lines 384-386) that the incorporation of ZnONPs reduces the toxicity, which is totally opposite to results shown in Figure 10.
19) First paragraph of conclusions sections does not refer to the article, just to general literature.
20) References format is not homogenous and some references are incomplete.
21) Weight and atomic percentages are not equal to 100% for any sample in Table S1.
Reviewer 3 Report
The article proposed by Authors is a well-organized and properly prepared work. The research topic is worth investigating. Discussion over the results of performed studies is correct. Final conclusions are promising. To sum up, the paper is worth considering for publication. Some minor improvements are suggested - all comments are given in a more detail below:
1) Manuscript contains some misspelling - including this one in the title (it should be "biocomposite" instead of "biocompozite") - which should be corrected.
2) Section 2.2.: information concerning the concentration of PVA solution used for the synthesis should be given.
3) Table 1: the subscript should be added to the notation of NH2 group.
4) Section 2.3.2.: in the description of the SEM methodology there is no information concerning the sputtering of samples. Was it applied?
5) Section 3.2.2.: Authors mentioned that "the hydrogels have become smoother when the amount of covalent crosslinker increases". Please, discuss the dependance between the surface morphology of hydrogels and the amount of crosslinker used in more detail.
6) Final conclusions should be given in a more quantified manner.
7) Reference [17] contains the whole journal name instead of its abbreviation.
Reviewer 4 Report
- The English structure needs to be modified, there are some grammatical mistakes in the manuscript.
- There are several papers with chitosan/PVA loaded with zinc oxide nanoparticles, what is the difference between your study and others? What is your novelty?
- The authors calculated the swelling ratio only after 24 hours, how did they make sure that this is the final values of the swelling ratio? The authors should measure the swelling ratios in different time intervals and when the amount of swelling is almost stable, it can be reported as final values. There maybe a chance of deswelling as well which can be realized only by measuring the swelling ration in different time intervals.
- Why did not the authors measure the tensile strength in presence of Zinc oxide nanoparticles? The final product that is going to be used is with zinc oxide, so, the mechanical properties of hydrogels with the nanoparticles is needed for comparison and conclusion.
- According to the MTT assay, adding ZNO reduced cell viability resulting in increasing toxicity but at the last paragraph, the authors mentioned that “The incorporation of ZnONPs into the hydrogel significantly reduced their toxicity” which is wrong, please modify this.
- It is better to measure the cell viability for another time internal (for example 48 hours) to make sure that cell viability will not decrease too much and will be stay in the reasonable range.
- How can the authors say that the effect of something is significant without statistical analysis? The statistical analysis should be calculated to realize if the effect is significant or not
Round 2
Reviewer 2 Report
The authors have addressed all of this reviewer's recommendations. The article could be accepted for publication in pharmaceuticsAuthor Response
Response to Reviewer 2 Comments
Point 1: The authors have addressed all of this reviewer's recommendations. The article could be accepted for publication in pharmaceutics
Response 1: Thanks for your appreciation.
Reviewer 4 Report
Some comments have addressed by the authors, however there are some more points to be modified before publication:
- There is no need to add statistical data table in the manuscript, mentioning to significant level is enough. You can add them in the supplementary data.
- Why did not the authors calculate the significant level for samples containing 3% ZnONPs?
- In the conclusion and abstract sections, the authors mentioned that “hydrogels are predominantly non-toxic or weak toxic” while hydrogels with 4 and 5% ZnONPs have moderate toxicity which should be mentioned, and this sentence should be corrected. Only hydrogels without ZnONPs were non-toxic, others were weak (only for3%) and moderate (for 4 & 5%) toxic
- How did the authors realize that the hydrogels are smoother by cross-section SEM pictures? You may realize the porosity by cross-section photos but for realizing the smoothness or being homogeneous, you need surface SEM photos
Round 3
Reviewer 4 Report
The authors addressed all the comments and the paper can be published.